# A Design of High-Efficiency: Vertical Accumulation Modulators Based on Silicon Photonics

**DOI:** 10.3390/nano13243157

**Published:** 2023-12-16

**Authors:** Zhipeng Zhou, Zean Li, Cheng Qiu, Yongyi Chen, Yingshuai Xu, Xunyu Zhang, Yiman Qiao, Yubing Wang, Lei Liang, Yuxin Lei, Yue Song, Peng Jia, Yugang Zeng, Li Qin, Yongqiang Ning, Lijun Wang

**Affiliations:** 1State Key Laboratory of Luminescence and Applications, Changchun Institute of Optics, Fine Mechanics and Physics, Chinese Academy of Sciences, Changchun 130033, China; zhouzhipeng21@mails.ucas.ac.cn (Z.Z.);; 2University of Chinese Academy of Sciences, Beijing 100049, China; 3Xiongan Innovation Institute, Chinese Academy of Sciences, Xiongan 071899, China; 4Jlight Semiconductor Technology Co., Ltd., Changchun 130033, China

**Keywords:** silicon photonics, plasma dispersion effect, optical modulator, modulation efficiency, loss-efficiency product

## Abstract

On-chip optical modulators, which are capable of converting electrical signals into optical signals, constitute the foundational components of photonic devices. Photonics modulators exhibiting high modulation efficiency and low insertion loss are highly sought after in numerous critical applications, such as optical phase steering, optical coherent imaging, and optical computing. This paper introduces a novel accumulation-type vertical modulator structure based on a silicon photonics platform. By incorporating a high-K dielectric layer of ZrO_2_, we have observed an increase in modulation efficiency while maintaining relatively low levels of modulation loss. Through meticulous study and optimization, the simulation results of the final device structure demonstrate a modulation efficiency of 0.16 V·cm, with a mere efficiency–loss product of 8.24 dB·V.

## 1. Introduction

Optical modulators are critical enablers in the fields of optical telecommunications, optical interconnects, optical sensing, and detection. Due to their low power consumption, high modulation bandwidth, rapid modulation speed, and mass-production capabilities, on-chip silicon photonic-based modulators have emerged as the most promising candidates, offering extraordinary solutions [1,2]. Their applications extend to optical linear matrix multiplications and optical phase steering, thereby necessitating densely deployed phase modulators [3,4,5]. Devices that demonstrate muscular modulation strength, high modulation efficiency, and low optical loss, as well as the potential for dense deployment with minimal crosstalk, are in high demand.

Generally, photonic modulators based on the thermo-optical effect can provide strong modulation with relatively low optical loss due to significant thermo-optical coefficients [6]. However, these thermal methods are less precise and suffer from significant crosstalk between channels, thereby hindering further advances in integration density. In contrast, electro-optic (EO)-based modulators offer precise optical phase control with minimal crosstalk, making them indispensable for large-scale, compact photonic circuits. The state-of-the-art silicon photonic EO modulators, based on plasma dispersion [7,8], exhibit the highest modulation strength. These operate with an accumulation-type structure, where carriers accumulate near the SiO_2_ dielectric layer between the N-doped and P-doped waveguides under an applied electric potential. However, the modulation efficiency of this approach remains insufficient, and optical loss is not negligible. The best-reported modulation performance is approximately 0.2 V·cm [9], with the efficiency–loss product limited to around 20 dB·V [10,11,12,13].

To address these issues, we propose a novel vertical structural accumulation-type silicon photonic modulator. Our approach, which differs from conventional capacitance accumulation modulators, introduces a dielectric layer of ZrO_2_ to replace SiO_2_. ZrO_2_, a high K (HK) material, exhibits a significantly larger dielectric constant than SiO_2_ [14,15], enhancing carrier accumulation capacity. Well-studied as an HK metal gate material in advanced MOSFET structures [16,17], ZrO_2_ can be easily incorporated using standard CMOS-compatible processes. Integrating ZrO_2_ into the design of carrier accumulation modulators improves modulation efficiency to 0.16 V·cm. Compared to traditional carrier accumulation modulators, our proposed design offers a 20% enhancement in modulation efficiency and a 40% decrease in the efficiency–loss product. This research addresses the challenges of low modulation efficiency and integration complexities in densely deployed modulators (Figure 1).

## 2. Principles and Methods

### 2.1. Principles

In silicon-based modulators, the plasmonic dispersion effect is the fundamental principle employed. Three primary modulation structures are utilized: (i) carrier injection-based [18], (ii) carrier depletion-based [19,20], and (iii) carrier accumulation-based [9,12,13]. In injection-type structures, modulation is attained by altering the semiconductor’s refractive index and absorption coefficient through electron and hole injection. These modulators are typically noted for their high modulation efficiency. However, their overall performance may be negatively influenced by the carrier lifetime limitations inherent in these systems. Depletion-based modulation, often achieved through reverse biasing of PN junctions, results in the highest modulation speed but suffers from a larger footprint, increased power consumption, and comparatively lower modulation efficiency.

In contrast, optical modulators based on carrier accumulation do not modulate carrier concentration through electrically controlled injection. Instead, they incorporate a dielectric layer within the waveguide, facilitating the accumulation of free carriers on either side of this layer. This design allows such devices to bypass the limitations imposed by minimal carrier lifetime, enhancing modulation speed. Conversely, accumulation-based modulation offers a balanced performance, achieving high modulation efficiency while optimizing speed, power consumption, and footprint [21].

When the wavelength is 1550 nm, the refractive index change is empirically related to the carrier concentration as follows [22]:(1)$$\Delta n=-5.40\times 10^{-22}N_{e}^{1.011}-1.53\times 10^{-18}N_{h}^{0.838}$$
(2)$$\Delta \alpha =8.88\times 10^{-21}N_{e}^{1.167}+5.84\times 10^{-20}N_{h}^{1.109}$$
where *N*_e_ represents the electron concentration, *N*_h_ represents the hole concentration, and the above is referred to as the Soref–Bennett formula.

The vertical structure is frequently regarded as one of the optimal choices for efficient modulators. Currently, this structure is predominantly constructed using Poly-Si-oxide-silicon. The peak modulation efficiency attained with this design is around 0.2 V·cm [9]. The key factor influencing modulation efficiency is the carrier accumulation capability, which is determined by material characteristics, such as (i) the dielectric constant of the oxide layer, (ii) the band gap in the oxide layer, and (iii) the affinity of electrons in the oxide layer. Of these, the dielectric constant plays the most crucial role. Therefore, employing high-K materials with a higher dielectric constant increases capacitance and enhanced carrier accumulation.

In the design of high-efficiency modulators that utilize carrier accumulation, three primary types are identified based on their gate electrode materials: ITO–oxide–silicon [23,24], III–V materials–oxide–silicon [25,26], and silicon–oxide–silicon [9,12,13]. Here, the choice of gate electrode material significantly influences the refractive index change. Modulators with ITO gate electrodes often exhibit high modulation efficiencies but suffer considerable modulation loss, limiting their efficiency benefits. Modulators using III–V gate electrode materials, while achieving higher efficiency and minimal losses, often require complex bonding operations in their fabrication, which pose challenges for large-scale integration. In contrast, silicon–oxide–silicon-based modulators, particularly those using SiO_2_ as the oxide layer, have reached the efficiency limits, prompting the need for innovative approaches to enhance their performance.

Figure 2 illustrates the design of the intended optical modulator using SOI (silicon-on-insulator) technology, which is compatible with CMOS processes. SOI technology is a prevalent methodology in fabricating silicon photonic chips and is extensively employed in producing various optoelectronic devices [27]. The design comprises three distinct layers: a top layer, a middle layer, and a bottom layer. These layers consist of polysilicon (poly-Si), an oxide layer, and crystal silicon. Both the poly-Si and the single-crystal silicon are doped with a concentration of 1 × 10^18^ cm^−3^ to reduce resistance, enhance the electro-optic (EO) bandwidth, and provide an abundance of free carriers for accumulation. The oxide layer, with a thickness of 5 nm, is made from zirconium dioxide (ZrO_2_). The poly-Si layer measures 150 nm thick, while the single-crystal silicon layer is 220 nm thick. The top electrode, which is positioned 900 nm from the optical field center, aligns with the bottom electrode, strategically placed along the left side of the single-crystal silicon. This configuration significantly mitigates losses, as indicated in reference [28].

### 2.2. Methods

The simulation work introduces the overall simulation tools, boundary conditions, grid division, and settings of the excitation port. This simulation adopts the finite element analysis method, which is divided into two parts: electrical simulation and optical simulation.

The electrical stimulation primarily involves applying different voltages at the excitation port to obtain varying distributions of carriers in semiconductor materials. The changes in the refractive index and absorption rate caused by the carrier variations are incorporated into the optical model for optical mode simulation using the Soref–Bennett formula. Except for the excitation port, all other boundaries are considered ideal insulators in the electrical simulation. The semiconductor material model is a steady-state model, which is calculated primarily from the following formula:(3)$$\nabla \cdot J_{n}=0$$
(4)$$\nabla \cdot J_{p}=0$$
(5)$$J_{n}=qn\mu_{n}\nabla E_{c}+qD_{n}\nabla n-qnD_{n}\nabla \ln\left(N_{c}\right)+qnD_{n,\;\mathrm{th}\;}\nabla \ln\left(T\right)$$
(6)$$J_{p}=qp\mu_{p}\nabla E_{v}-qD_{p}\nabla p+qpD_{p}\nabla \ln\left(N_{v}\right)-qpD_{p,\;\mathrm{th}\;}\nabla \ln\left(T\right)$$
in the optical simulation, the working wavelength is set to 1550 nm. A comprehensive optical mode simulation is conducted after introducing the refractive index changes caused by the carriers into the optical model. The boundary conditions for this simulation are ideal conductors. Since this is an optical field mode analysis simulation, no fixed excitation exists. The primary formula used during the optical mode simulation process is:(7)$$\nabla \times \left(\nabla \times E\right)-k_{0}^{2}\epsilon_{r}E=0$$
(8)$$\alpha =j\beta +\delta_{z}=-\lambda$$
(9)$$E\left(x,y,z\right)=\overset{\sim }{E}\left(x,y\right)e^{-\alpha z}$$
finally, the grid division is consistent in both the electrical and optical simulations, utilizing a grid size with a side length of 1 nanometer (nm). This acceptable grid density helps to enhance the reliability of the simulation results.

Moreover, in designing the optical modulator, several key parameters are prioritized as our primary criteria for design and optimization. First, the magnitude of the applied voltage is critical, as it directly affects compatibility with the standard voltage used in the CMOS-process-integrated circuits. This parameter is essential for enhancing the density of the optoelectronic co-design. Second, the modulation efficiency, represented as VπL (V·cm), the optical loss (measured in dB/cm), and the optimal figure of merit (FOM), which combines specific values to balance both aspects, are central to our evaluation. The lower the numerical value for the modulation efficiency, the higher the integration capability of the modulator. Similarly, a lower efficiency–loss product (αVπL) indicates greater cost-effectiveness in the design context.

## 3. Results

To assess the differences in carrier accumulation capacities between high K (HK) materials and conventional SiO_2_ dielectric materials, we conducted a comparative analysis under a −2 V voltage condition for materials. Figure 3 illustrates that using HK material significantly increases carrier accumulation on both sides of the dielectric layer. Based on empirical formulas, this marked increase will lead to enhanced modulation efficiency and corresponding changes in the effective mode refractive index.

Having demonstrated the superior carrier accumulation capabilities of high-K (HK) materials, our research next focused on exploring the relationship between applied voltage and modulation performance. The range of applied electrode voltages spanned from −4 V to 0 V. Our simulations, illustrated in Figure 4, show a consistent decrease in the VπL value as voltage increases. Notably, at −1.4 V, a critical juncture is observed where α·VπL reaches an optimal point. At this voltage, the modulation efficiency peaks at 0.29 V·cm, accompanied by a relatively low optical loss of 38.31 dB/cm and an efficiency–loss product of 11.12 dB·V. These metrics significantly outperform those of traditional silicon-based SiO_2_ dielectric materials that are typically used in modulator devices. Capitalizing on this critical voltage point opens avenues for further performance enhancement through structural optimizations.

Three principal factors determine the modulation efficiency of optical modulators. Firstly, the characteristics of the dielectric layer, which include intrinsic properties, such as dielectric constant, electron affinity, and bandgap width, are critical, as well as dimensional attributes, including layer thickness. These factors directly influence the capacitor’s value and the capacity for carrier accumulation. Secondly, doping concentration significantly impacts modulation efficiency, especially regarding carrier density in monocrystalline and poly-Si. Lastly, there is a notable correlation between the modulator’s geometric variables, such as the thickness and etching depth of poly-Si, and its modulation efficiency. Collectively, these factors substantially impact the modulation efficiency, optical loss, and efficiency–loss product of the modulator.

Our approach to designing and optimizing modulators is meticulously structured into three distinct phases, with each phase focused on enhancing the performance aspects mentioned above. Through this phased methodology, we aim to enhance the modulator’s performance by precisely adjusting these variables. This approach aims to achieve higher modulation efficiency and the lowest possible product of loss–efficiency.

### 3.1. Dielectric Layer Thickness

Firstly, the impact of the dielectric layer thickness is approached by defining the planar capacitor formula:(10)$$C=A\frac{\varepsilon_{0}\varepsilon_{r}}{d}$$
where A denotes the planar surface area, d represents the distance between the two electrode plates, $\varepsilon_{0}$ stands for the vacuum permittivity, and $\varepsilon_{r}$ is the dielectric constant of the oxide layer material.

According to the formula, an increase in the oxide layer thickness leads to a reduction in capacitance, which, in turn, diminishes the capacity for carrier accumulation. To investigate the relationship between the thickness of the oxide layer and modulation performance, we generated performance graphs for each modulator with various oxide layer thicknesses, from 5 to 11 nanometers, and under different voltage conditions. The results of these tests are depicted in Figure 5a,b.

The observed outcomes align with the predictions made by the capacitance formula. As the thickness of the dielectric layer increases, capacitance decreases, resulting in reduced carrier accumulation. When evaluating the modulation efficiency and efficiency–loss product parameters, using thicker oxide layers than the optimized voltage range proves counterproductive. Such thicknesses lead to a higher efficiency–loss product and lower modulation efficiency. Considering the complexities inherent in manufacturing processes, the optimal thickness of the oxide layer was 5 nm.

### 3.2. Doping Concentration

The next step in our study involved evaluating the impact of doping, a critical parameter that directly affects modulation capabilities. Higher doping concentrations introduce a more significant number of carriers, thereby enhancing modulation abilities. However, empirical formulas indicate that holes, compared to electrons, result in more considerable refractive index changes and reduced losses. To confirm this observation, we analyzed the variations in losses and efficiency–loss product at different concentrations of P-doping and N-doping, specifically at the critical voltage of −1.4 V. The results of this analysis are presented in Figure 6a,b.

Both P-doping and N-doping demonstrate increased losses. However, lower losses and efficiency–loss products are attainable when reducing the P-doping concentration. At a P-doping concentration of 2 × 10^17^ 1/cm^3^, the losses were merely at 20.46 dB, resulting in an exceedingly low efficiency–loss product of 6.19 dB·V. In contrast, within the same range of concentration variations, the minimum efficiency–loss product for N-doping was confined to 9.5 dB·V. Therefore, our data suggest that minimizing P-doping near the pole voltage in our designed structure can yield superior modulation performance.

Moreover, according to empirical formulas, holes demonstrate lower absorption than electrons. However, they generate more significant changes in the refractive index. Consequently, maintaining hole concentrations that are too low might result in a decrease in modulation capabilities.

### 3.3. Modulator Geometric Factors

The last crucial factor affecting modulation efficiency is the influence of geometry. The distance between the region where carriers amass high charge density, and the center of the optical mode considerably affects both the modulation efficiency and the product of the loss–efficiency. The highest modulation efficiency occurs when these elements perfectly align, as suggested by the following formula:(11)$$n_{eff}\left(V\right)=n_{eff,i}+\frac{\int E^{*}\left(y\right)\cdot \Delta n\left(y,V\right)E\left(y\right)dy}{\int E^{*}\left(y\right)\cdot E\left(y\right)dy}\cdot \frac{dn_{eff}}{dn_{co}}$$
here, the equation denotes $n_{eff,i}$ as the effective refractive index of the non-doped optical waveguide, d*n*_eff_/d*n*_co_ as the ratio of the mode change effective refractive index to the change in refractive index of the optical waveguide core layer, which is very close to 1, and *E*(*y*) as the 1D electric field distribution obtained via the effective refractive index method.

The primary influencers of the modulator’s ultimate performance are derived from the formula mentioned above and the current geometric configuration. The key aspects are predominantly the ridge height (aligned with the thickness of the poly-Si) and the depth of etching (relating to the alteration in the refractive index within the single-crystal silicon region).

We initiated the investigation by analyzing the etching depth. Subsequently, based on our model configuration and the manufacturing process, we can convert the silicon on both sides of the waveguide into silicon dioxide through an etching operation. Therefore, we can assume an etching depth of 155 nm at the initial state, as depicted in Model Figure 2. Here, the sides of the poly-Si are entirely composed of silicon dioxide. However, in this research, we investigated the impact of silicon dioxide confinement on the overall modulation performance by increasing the etching depth to transform the monocrystalline silicon region into silicon dioxide. The results are illustrated in Figure 7a,b.

An increased etching depth facilitates the transformation of single-crystal silicon adjacent to the center of the optical field into silicon dioxide. This change alters the refractive index, resulting in a more concentrated electric field intensity on both sides of the dielectric layer. Our graph directly correlates with increased etching depth and enhanced modulation efficiency. Thus, we analyzed a normalized light field intensity profile along line AB to support this hypothesis, overlaid with carrier concentration data at a voltage of 4 V. This analysis is visually represented in Figure 8, where it is evident that a greater etching depth is associated with higher normalized light field intensity values on both sides of the dielectric layer, leading to increased modulation efficiency.

In our investigation of the efficiency–loss product, we performed scans of this parameter at various etching depths using four different voltage settings. A clear and consistent pattern emerged from this analysis: At all voltage levels, the efficiency–loss product decreased as the etching depth increased. This finding confirms that augmenting the etching depth effectively reduces the loss–efficiency product, thus, improving the modulator’s performance.

Another critical factor causing modulation performance variations is the thickness of the poly-Si. Here, changes in poly-Si thickness led to alterations in the thickness of the waveguide, which in turn impacted the central position of the optical field. Similar to the influence of the etching depth, this effect can be attributed to the degree of overlap between the center of the optical field and the region of the dense carrier accumulation. We conducted a performance scan on the modulator using a range of poly-Si thicknesses to elicit clear and intuitive conclusions. Since the initial poly-Si thickness was 150 nm, we set the scanning range from 120 to 200 nm. The outcomes of this scan are presented in Figure 9a,b.

Increasing the poly-Si thickness impacts modulation efficiency while concurrently decreasing the efficiency–loss product. As the poly-Si thickness increases, the normalized optical field strength on both sides of the dielectric layer also rises, thus, boosting the modulation efficiency. However, this increase in thickness has its constraints. Maintaining the overall polysilicon thickness below 220 nm is crucial to meet the single-mode optical transmission condition. Notably, and irrespective of the applied voltage, the trend toward a product with reduced efficiency–loss and increased polysilicon thickness remains consistent, providing a valuable guideline for our optimization design process.

To more intuitively demonstrate the influence of geometric factors on modulation efficiency, we have developed optical field intensity diagrams under varied geometric scenarios, presented in Figure 10a,b. Indeed, Figure 10a illustrates the optical field intensity distribution for an etching depth of 155 nm and a polysilicon thickness of 150 nm. In this configuration, the central position of the optical field intensity is predominantly concentrated in the center of the monocrystalline silicon layer, considerably distant from the sides of the dielectric layer. Conversely, Figure 10b shows a noticeable upward shift in the central position of the optical field when the etching depth and the polysilicon thickness were increased to 370 nm and 200 nm, respectively. This shift increased the overlap with the dielectric layer’s position, resulting in significantly higher modulation efficiency. These observations confirm our proposed hypothesis and underscore the crucial role of geometric modifications and the complex relationship between physical structure and functional efficacy in photonic devices.

Building upon insights from our previous research, we redesigned and optimized the structure of our modulator. The outcomes of this re-simulation are depicted in Figure 11. After optimization, the modulator achieves its peak efficiency–loss product at a voltage of −1.2 V—registering a remarkable 6.1 dB·V. This peak corresponds to a modulation efficiency of 0.26 V·cm. It is important to note that these results specifically pertain to the efficiency–loss product. However, increasing the applied voltage is a viable strategy should a higher modulation efficiency be the priority. For example, at −4 V, the modulator can surpass a modulation efficiency of 0.1 V·cm, attaining an impressive 0.096 V·cm.

Nevertheless, a working voltage of −1.6 V appears optimal following a comprehensive evaluation. At this voltage, the modulator exhibits a modulation efficiency of 0.16 V·cm and a corresponding efficiency–loss product of 8.24 dB·V. This optimization process showcases the modulator’s efficiency across different electrical conditions and underscores the critical need to balance the modulation efficiency and efficiency–loss product for optimal performance in practical applications.

Upon the foundation of our optimized structure, we conducted a comparative analysis of the simulation results for non-optimized structures using SiO_2_ and ZrO_2_ dielectric layers, as shown in Figure 12. The collected data distinctly highlights the advantages of using high-k (HK) materials in modulators. Whether considering modulation capability or efficiency–loss metrics, modulators employing ZrO_2_, an HK material, consistently surpass those with traditional dielectric materials. Furthermore, the optimized structure demonstrates significant performance enhancements compared to the initial design. This improvement confirms the effectiveness of the structural optimization. In a side-by-side comparison using the same framework, our optimized structure, when contrasted with traditional SiO_2_ dielectric layers, shows a clear and substantial increase in the simulation results. Both the efficiency–loss product and modulation efficiency increased by over 200%.

One of the primary metrics for evaluating the efficiency of optical modulators is the phase shift length. Our study compared the phase shift lengths of optical modulators employing traditional SiO_2_ dielectric layers, high-k (HK) material ZrO_2_ dielectric layers, and optimized ZrO_2_ dielectric layers through simulations. As depicted in Figure 13, the optical modulators with ZrO_2_ dielectric layers demonstrate an approximate 200% improvement in phase shift length compared to those with traditional layers. Following the optimization of the device’s geometric structure and doping, the phase shift length was further reduced based on these findings. Specifically, at a voltage of −1.4 V, the phase shift length was 1.14 mm, and at voltages below −1.6 V, it was less than 1 mm. This reduction in phase shift length contributes to greater integration, underscoring the benefits of using HK material ZrO_2_ dielectric layers in optical modulators. 

In our final simulation, we assessed the leakage current and bandwidth of the optical modulators with optimized ZrO_2_ dielectric layers. Figure 14a illustrates that the leakage current consistently stays below 7.6 × 10^−5^ A/cm^2^ at voltages above −3.2 V. However, at voltages below −3.2 V, there is a marked increase in leakage current, culminating in a peak of 0.039 A/cm^2^ at −4.0 V.

An analysis of the device’s S11 parameter indicates a −3 dB bandwidth of 2.94 GHz, as depicted in Figure 14b. The utilization of high-k (HK) materials increases overall capacitance compared to traditional SiO_2_ dielectric layers. Although the bandwidth of optical modulators using HK material dielectric layers still necessitates further optimization—especially when contrasted with traditional SiO_2_ layer modulators, which can achieve bandwidths exceeding 10 GHz—this aspect will be a primary focus in our future research endeavors.

## 4. Discussion

High-efficiency modulators often incorporate accumulation capacitors in their design. The capacity of the oxide layer to accumulate carriers is crucial in determining modulation efficiency. This efficiency is also influenced by the doping concentration, the modulator’s geometry, and the applied voltage. Geometric factors are instrumental in dictating carrier accumulation and its interaction with the optical field distribution, which, in turn, impacts both modulation efficiency and losses. An increased overlap between these elements results in higher modulation efficiency, leading to more significant losses. On the other hand, a thicker oxide layer diminishes capacitance and carrier accumulation capability, which reduces modulation efficiency.

Regarding the effects of doping, modulation capability is closely linked to carrier concentration. However, N/P doping scenarios exhibit distinct behaviors. Hence, increasing doping concentrations in the N and P-type regions can enhance modulation efficiency, yet the resulting losses vary significantly. At equivalent doping concentrations, holes demonstrate more pronounced modulation efficiencies, leading to reduced losses and a lower efficiency–loss product. Thus, strategically employing holes as the primary modulation carriers is a crucial method for improving the efficiency–loss product.

This study suggests using a vertical structure capacitor modulator constructed with an innovative oxide layer material to achieve enhanced modulation efficiency and reduced efficiency–loss products, while also simplifying the manufacturing process. When focusing on a single performance goal, we can attain an ultra-low efficiency–loss product of 6.1 dB·V and an exceptional modulation efficiency of 0.096 V·cm. Nevertheless, this involves a trade-off with losses. The realized modulation efficiency is 0.16 V·cm, corresponding to an efficiency–loss product of 8.24 dB·V. A comparison with other silicon-based accumulation modulators is provided in Table 1.

**Table 1 nanomaterials-13-03157-t001:** Performance parameters of accumulation modulators with different gate materials.

Gate Material	Oxide	VπL (V·cm)	Loss (dB/cm)	α· VπL (dB·V)	Result Type
ITO [23]	Al_2_O_3_	0.052	>1500	>80	Experimental
ITO [24]	Al_2_O_3_	0.095	16,000	152	Experimental
InGaAsP [25]	Al_2_O_3_	0.047	4.6	<1	Experimental
InP [26]	SiO_2_/Al_2_O_3_	0.54	2.3	1.24	Experimental
Poly-Si [7]	SiO_2_	1.8	>30	>48	Experimental
Poly-Si [29]	SiO_2_	0.886	>18	>16	Numerical
Poly-Si [9]	SiO_2_	0.2	65	13	Experimental
(This work) Poly-si	ZrO_2_	0.16	50	8.24	Numerical

Compared to traditional SiO_2_ dielectric layer capacitor structures, our modulator demonstrated significant advancements in modulation capabilities. Our approach has resulted in a 20% increase in modulation capacity and a notable 40% decrease in the efficiency–loss product, ultimately, outperforming the most advanced high-efficiency modulators currently available.

Our modulator also shows substantial improvements over alternative approaches, such as those using ITO gate materials. Specifically, we have achieved a significant reduction in modulation losses. Moreover, compared to III–V material modulators, our design eliminates the need for additional bonding processes, thereby simplifying the overall complexity of the manufacturing procedure. This advancement creates new possibilities for future research and offers innovative solutions for achieving high-density, high-efficiency, and high-precision modulations.

## 5. Conclusions

This paper presents a high-density modulator based on a vertical structure incorporating CMOS-compatible high-k (HK) material dielectric layers, featuring an innovative accumulation structure using ZrO_2_ oxide layers. Our simulations indicate that the modulator achieves a modulation efficiency of 0.16 V·cm in the vertical direction, with a corresponding loss-efficiency product of 8.24 dB·V. Further optimization of the voltage settings could lead to even more impressive modulation efficiency, potentially lowering the efficiency–loss product to 6.1 dB·V and fulfilling the requirements for high-precision, dense, and efficient modulations.

In summary, our research underscores the advantages of HK materials in constructing high-efficiency optical modulators. The significant benefits of utilizing high-k materials, such as ZrO_2_, and structural optimization have opened new avenues for enhancing device performances. This breakthrough emphasizes the critical role of material selection and structural design in optimizing the optical modulator efficiency. It establishes a new standard in the field, fostering further research and innovation in photonic technology. This advancement heralds exciting prospects for developing more efficient, powerful, and versatile optical communication systems. However, further research is essential to evaluate its performance and explore its practical applications fully. Nevertheless, the outcomes of this research mark a significant step forward in devising innovative designs for high-efficiency optical modulators.

## Figures and Tables

**Figure 1 nanomaterials-13-03157-f001:**
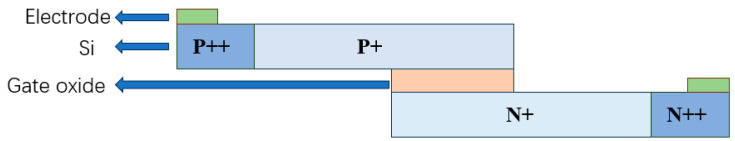
Schematic diagram of conventional accumulation-type electro-optic modulator.

**Figure 2 nanomaterials-13-03157-f002:**
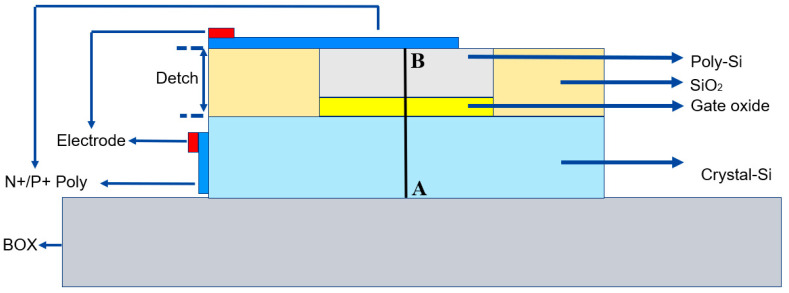
Schematic diagram of the initial design model.

**Figure 3 nanomaterials-13-03157-f003:**
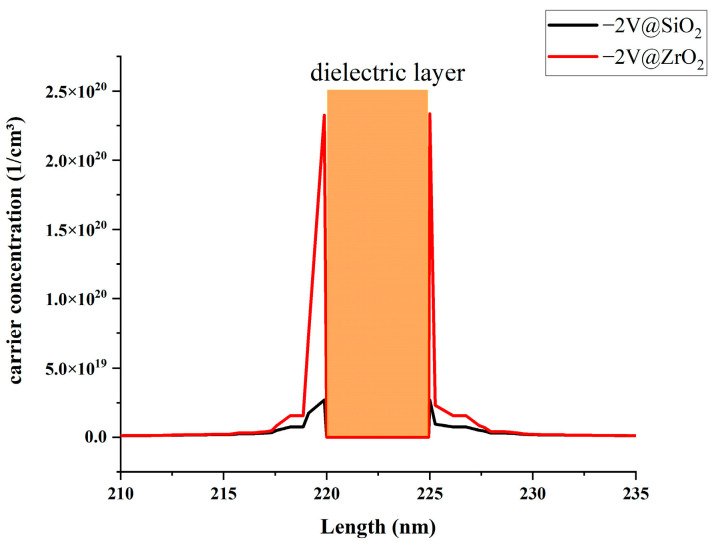
Traditional material vs. HK material carrier accumulation comparison.

**Figure 4 nanomaterials-13-03157-f004:**
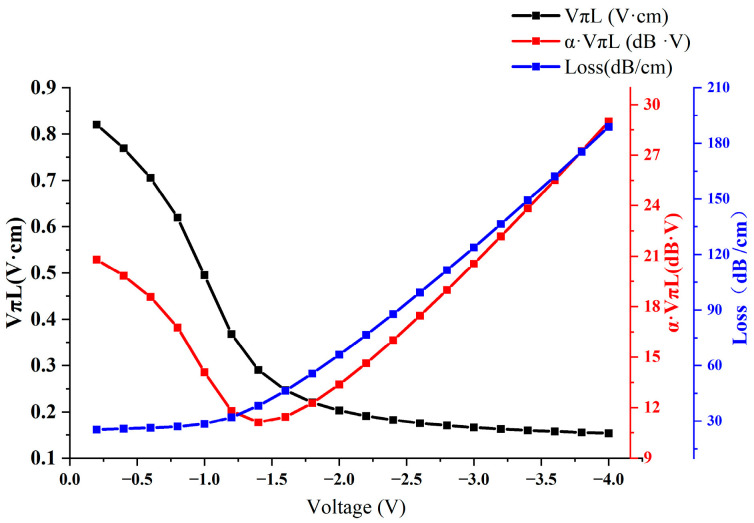
Voltage–modulation performance relationship chart.

**Figure 5 nanomaterials-13-03157-f005:**
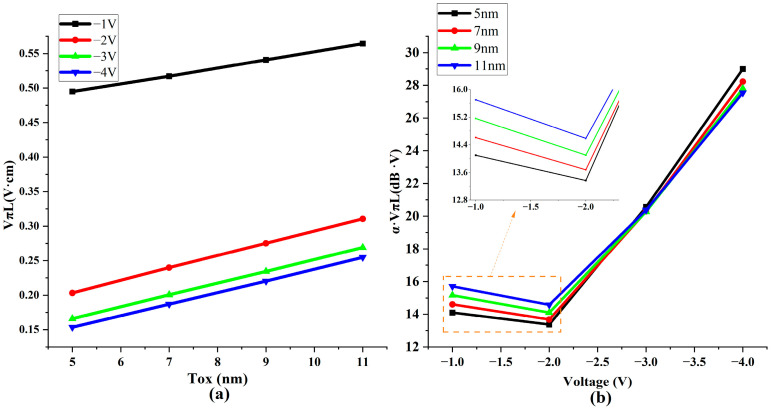
Relationship between oxide layer thickness (Tox) and modulation performance. (**a**) Schematic diagram of modulation efficiency changing with oxide layer thickness (**b**) Efficiency–loss product changing with voltage and oxide layer thickness.

**Figure 6 nanomaterials-13-03157-f006:**
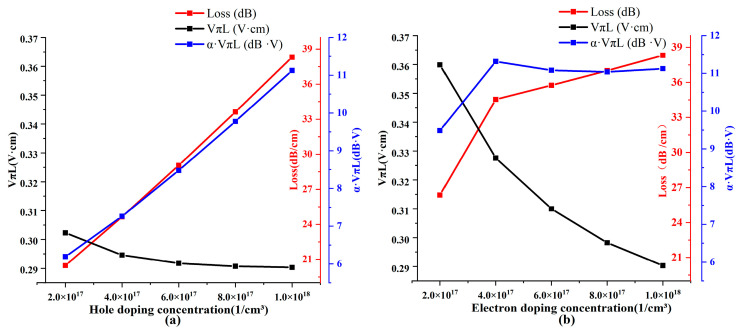
Performance variation under different doping conditions. (**a**) P-doping and (**b**) N-doping.

**Figure 7 nanomaterials-13-03157-f007:**
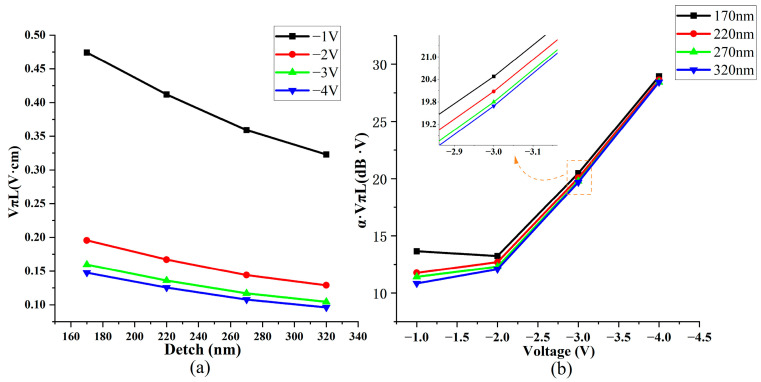
Schematic of modulation performance at different etching depths (Detch). (**a**) Schematic diagram of modulation efficiency changing with etching depth and voltage. (**b**) Loss efficiency product changing with voltage and etching depth.

**Figure 8 nanomaterials-13-03157-f008:**
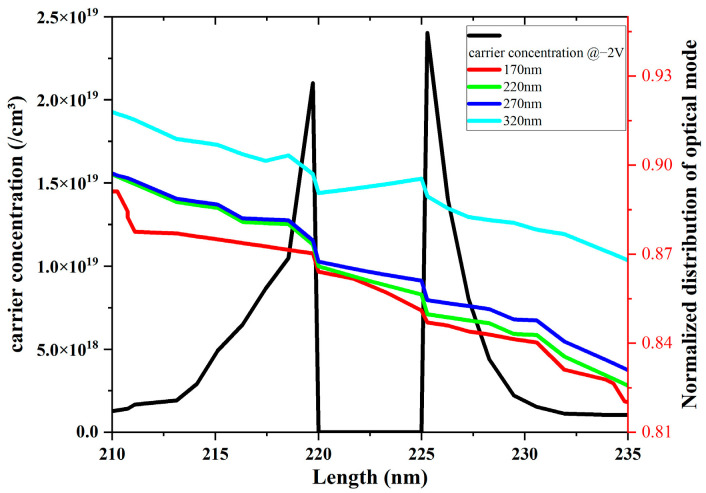
Schematic of overlapping normalized electric field intensity with carriers.

**Figure 9 nanomaterials-13-03157-f009:**
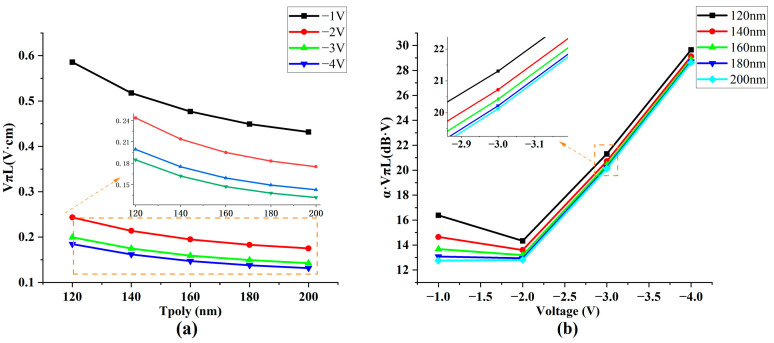
Schematic of performance with different poly-Si thicknesses (T poly). (**a**) Schematic diagram of modulation efficiency changing with poly-Si thickness (**b**) Efficiency–loss product changing with voltage and poly-Si thickness.

**Figure 10 nanomaterials-13-03157-f010:**
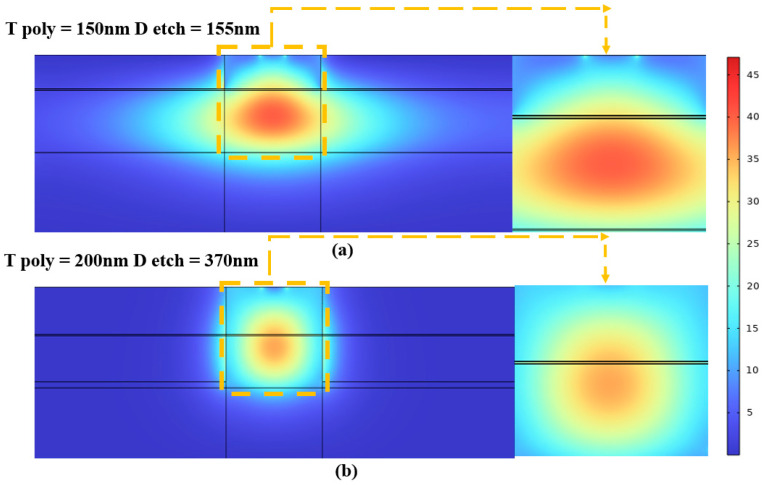
Schematic of optical field distribution under different geometric conditions (**a**) T poly = 150 nm, D etch = 155 nm and (**b**) T poly = 200 nm, D etch = 370 nm.

**Figure 11 nanomaterials-13-03157-f011:**
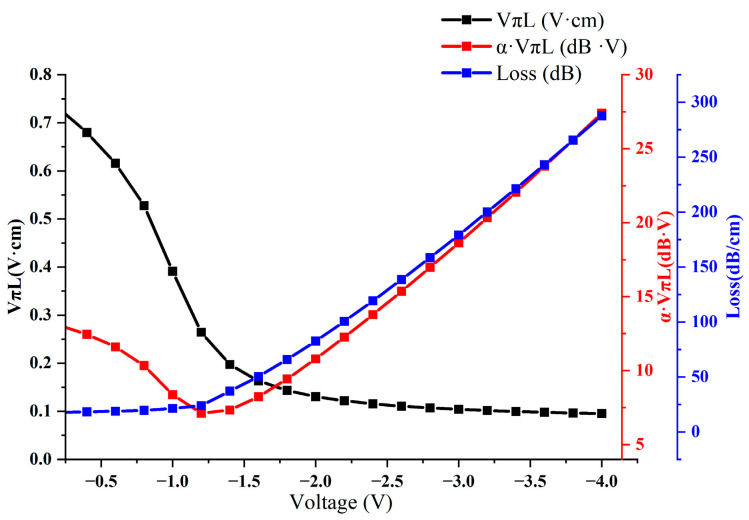
Schematic of modulator performance after optimization.

**Figure 12 nanomaterials-13-03157-f012:**
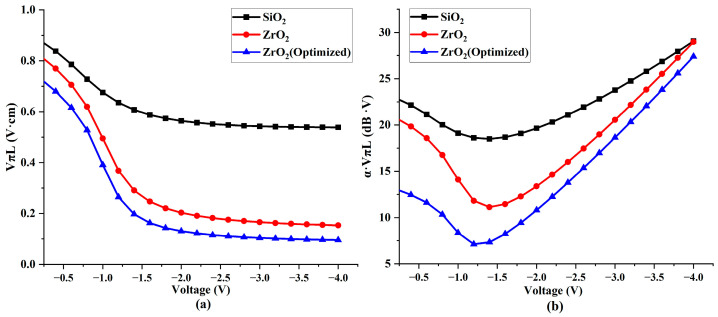
Comparison chart of modulator performances after optimization (**a**) Comparison chart of modulation efficiencies of different materials. (**b**) Comparison chart of efficiency–loss products using different materials.

**Figure 13 nanomaterials-13-03157-f013:**
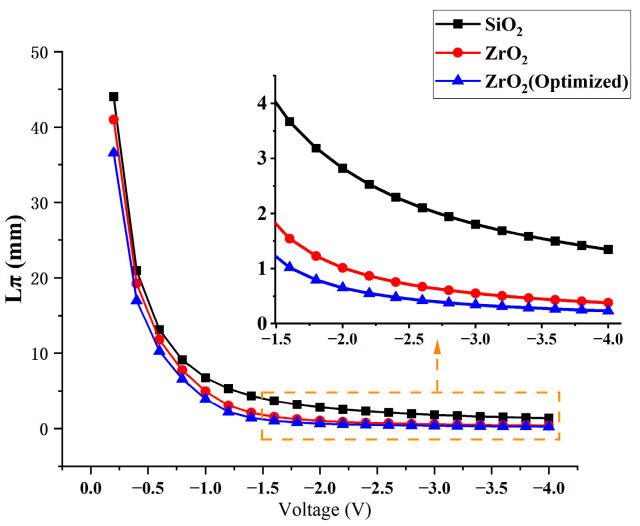
Schematic illustration of phase shift length variation with voltage.

**Figure 14 nanomaterials-13-03157-f014:**
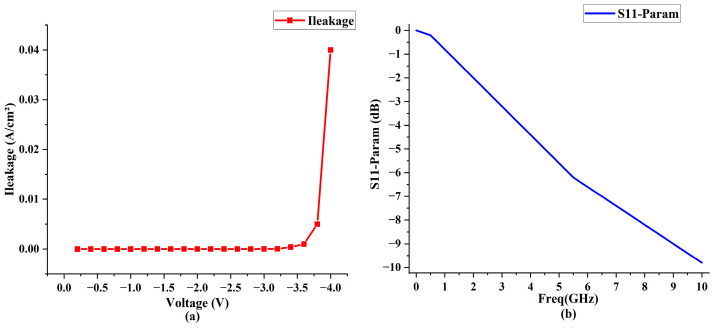
Simulation graphs of leakage current and bandwidth. (**a**) Schematic illustration of leakage current variation with voltage. (**b**) Schematic illustration of bandwidth based on -S11 parameter.

## Data Availability

The data presented in this study are available on the request from the corresponding author.

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
