# Peer review of "A Design of High-Efficiency: Vertical Accumulation Modulators Based on Silicon Photonics"

_nanomaterials, 2023, doi:10.3390/nano13243157_

Round 1

Reviewer 1 Report

Comments and Suggestions for Authors

I have the following suggestions

1) The author said "Depicted in Figure 2 is the initial design model structured as a rib waveguide, fabri- 110 cated through the SOI (Silicon on Insulator) process. " What does that mean? Do you mean CMOS fabrication process? Make correction. 

2) In figure 5 (a) what is "tox" on x-axis? Write the figure labels as (a) and (b) instead of a and b. (I mean in brackets). Correct all the figures.

3) Whay colon at the end of the sentence? "The resulting outcomes are visualized in Figures 5(a) and (b): "

4) What is the on the Y-axis of Figure 6, figure 11? Label it.

5) In table 1, it should be mentioned which results are experimental and which are numerical-based studies. 

6) The current form of the paper has a lot of grammatical mistakes and the sentence formation is very poor. I suggest revising the manuscript with the help of a native English speaker. 

7) The paper is based on numerical results, yet there is no detail on the simulation tool, boundary conditions, meshing, excitation port, etc. I suggest adding a section on the numerical model. 

Comments on the Quality of English Language

Very poor. Needs extensive English correction. 

Author Response

Dear Editors and reviewers:

Thanks a lot for your reply and comments concerning our manuscript entitled “A design of high-efficiency, vertical accumulation modulator based on silicon photonics”(Article, No. nanomaterials-2743558). We appreciate your patience reviewing our manuscript as well as giving professional and meticulous advises. Those comments are constructive and inspiring. Some of the suggestions have even provided profound insights and guidance for our research work. We believe that the quality of our article is going to be significantly improved followed by the valuable guidance. We have studied every comment carefully and have made correction which we hope meet with approval. Revised portion are marked in red in the paper. The main corrections in the paper and the responds to the reviewers' comments are as following:
Comment 1

1.The author said "Depicted in Figure 2 is the initial design model structured as a rib waveguide, fabri- 110 cated through the SOI (Silicon on Insulator) process. " What does that mean? Do you mean CMOS fabrication process? Make correction. 

Response 11.First, we thank you for taking the time to review our manuscript. Your insightful questions made us realize that our description of the manufacturing process was unclear. To address your concerns, we have revised this section, as shown from line 110 to 113 on page 3. We appreciate your attention to detail, which helps improve the quality of our manuscript.

The revised content are as follows:

Figure 2 illustrates the design of the intended optical modulator, constructed using SOI (Silicon-on-Insulator) technology that is compatible with CMOS processes. SOI technology is a prevalent methodology in the fabrication of silicon photonic chips, extensively employed in the production of various optoelectronic devices[27].

Comment 2

2.In figure 5 (a) what is "tox" on x-axis? Write the figure labels as (a) and (b) instead of a and b. (I mean in brackets). Correct all the figures.

Response 2

2.We sincerely thank you for the time and effort you have invested in evaluating our manuscript. We apologize for the oversight that led to the omission of annotations for the horizontal axis (Tox) in Figure 5(a), and we have carefully reviewed all figures for abbreviations like Tpoly, Detch, etc., and provided explanations in the captions. Realizing the importance of accurate picture labels for maintaining the rigor of the paper, we have checked and amended all figures as required.

Comment 3

 3.Whay colon at the end of the sentence? "The resulting outcomes are visualized in Figures 5(a) and (b): "

Response 33.Thank you for your thorough and detailed review of our manuscript. Correct punctuation is essential for the rigor of academic writing, and we have carefully re-examined and corrected our use of punctuation marks.
Comment 4

  1. What is the on the Y-axis of Figure 6, figure 11? Label it.

Response 4
4.We deeply apologize for the oversights during the review process, which led to several unintended errors. We have meticulously reviewed and corrected the coordinate annotations in all figures.Comment 5

5.In table 1, it should be mentioned which results are experimental and which are numerical-based studies. 

Response 5

5.Thank you for pointing out this important issue, which is essential for academic rigor. We have now included distinctions in Table 1.

The revised content are as follows:

Gate material

Oxide

VπL(V·cm)

Loss(dB/cm)

α· VπL (dB·V)

Type of Result

ITO[23]

Al2O3

0.052

>1500

>80

Experimental

ITO[24]

Al2O3

0.095

16000

152

Experimental

InGaAsP[25]

Al2O3

0.047

4.6

<1

Experimental

InP[26]

SiO2/Al2O3

0.54

2.3

1.24

Experimental

Poly-Si[7]

SiO2

1.8

>30

>48

Experimental

Poly-Si[29]

SiO2

0.886

>18

>16

 Numerical

Poly-Si[9]

SiO2

0.2

65

13

Experimental

(This work)

Poly-si

ZrO2

0.16

50

8.24

 Numerical

 Comment 6

6.The current form of the paper has a lot of grammatical mistakes and the sentence formation is very poor. I suggest revising the manuscript with the help of a native English speaker. 

Response 6
6.We appreciate your valuable comments. I have rechecked and revised the grammatical errors and reorganized the sentence formation as per your requirements. We apologize for the previous issues in expression and earnestly request your understanding.

Comment 7

7.The paper is based on numerical results, yet there is no detail on the simulation tool, boundary conditions, meshing, excitation port, etc. I suggest adding a section on the numerical model. 

Response 7

7.Thank you for your insightful comments and valuable suggestions. The added sections not only enrich the content of our article but also enhance its academic depth and breadth, making it more targeted and practical. From lines 125 to 150 on page 3, we introduced the methods used in the simulations, boundary conditions, grid division, excitation conditions, and some of the simulation principle equations involved.

The revised content are as follows:

2-2:Methods

The simulation work introduces the overall simulation tools, boundary conditions, grid division, and settings of the excitation port, etc. This simulation adopts the finite element analysis method, which is divided into two parts: electrical simulation and optical simulation.

    The electrical simulation primarily involves applying different voltages at the excitation port to obtain varying distributions of carriers in semiconductor materials. The changes in refractive index and absorption rate caused by the carrier variations are incorporated into the optical model for optical mode simulation using Soref-Bennet formula. In the electrical simulation, except for the excitation port, all other boundaries are considered as ideal insulators. The semiconductor material model is a steady-state model, primarily using the following formula:

In the optical simulation, the working wavelength is set to 1550 nm. After introducing the refractive index changes caused by the carriers into the optical model, a comprehensive optical mode simulation is conducted. The boundary conditions for this simulation are ideal conductors. Since this is a simulation for optical field mode analysis, there is no fixed excitation. The primary formula used during the optical mode simulation process is:

Finally, the grid division is consistent in both electrical and optical simulations, utilizing a grid size with a side length of 1 nanometer (nm). This acceptable grid density helps to enhance the reliability of the simulation results.

Reviewer 2 Report

Comments and Suggestions for Authors

This paper demonstrates the design of the high-efficiency, vertical accumulation modulator based on silicon photonics. Overall, the manuscript is well written. The paper can be accepted after a major revision. Some issues should be revised:

1.      For the VpL in Fig. 4, could the authors provide the length of the device? That is because the device size is an important issue for future Silicon photonics applications.

2.      For this device design, do the authors evaluate the leakage current?

3.      Besides the VpL, could the authors provide the simulated device bandwidth?

Author Response

Dear Editors and reviewers:

Thanks a lot for your reply and comments concerning our manuscript entitled “A design of high-efficiency, vertical accumulation modulator based on silicon photonics”(Article, No. nanomaterials-2743558). We appreciate your patience reviewing our manuscript as well as giving professional and meticulous advises. Those comments are constructive and inspiring. Some of the suggestions have even provided profound insights and guidance for our research work. We believe that the quality of our article is going to be significantly improved followed by the valuable guidance. We have studied every comment carefully and have made correction which we hope meet with approval. Revised portion are marked in red in the paper. The main corrections in the paper and the responds to the reviewers' comments are as following:

Comment 1

1.For the VpL in Fig. 4, could the authors provide the length of the device? That is because the device size is an important issue for future Silicon photonics applications.

Response 1

1: Thank you for thoroughly reading our manuscript and for your constructive suggestions. In response to your advice on modifying the length of the optical modulator, we have added detailed illustrations for devices with different lengths under three scenarios: using SiO2 dielectric layer, using ZrO2 dielectric layer, and using optimized ZrO2 dielectric layer. These details are provided from lines 362 to 372 on page 11 and in Figure 13. We greatly appreciate your keen eye for detail, which helps enhance the quality of our manuscript.

The added content is as follows:

One of the primary metrics for evaluating the efficiency of optical modulators is the phase shift length. Our study compared the phase shift lengths of optical modulators employing traditional SiOâ‚‚ dielectric layers, high-k (HK) material ZrOâ‚‚ dielectric layers, and optimized ZrOâ‚‚ dielectric layers through simulations. As depicted in Figure 13, the optical modulators with ZrOâ‚‚ dielectric layers demonstrate an approximate 200% improvement in phase shift length compared to those with traditional layers. Following the optimization of the device's geometric structure and doping, the phase shift length was further reduced based on these findings. Specifically, at a voltage of -1.4V, the phase shift length is 1.14mm, and at voltages below -1.6V, it is less than 1mm. This reduction in phase shift length contributes to greater integration, underscoring the benefits of using HK material ZrOâ‚‚ dielectric layers in optical modulators.

Figure 13: Schematic Illustration of Phase Shift Length Variation with Voltage

Comment 2

2.For this device design, do the authors evaluate the leakage current?

Response 22: We greatly appreciate your valuable comments and suggestions, which not only enrich the content of our article but also enhance its academic depth and breadth, making it more targeted and practical. Here, we evaluated the leakage current density of the modulator using a ZrO2 dielectric layer post-optimization, and this is described from lines 378 to 383 on page 12 and in Figure 14(a). Again, thank you for your valuable advice, which has highlighted the importance of leakage current and provided new evaluation points for subsequent accumulative optical modulator designs. The added content is as follows:

In our final simulation, we assessed optical modulators' leakage current and bandwidth with optimized ZrOâ‚‚ dielectric layers. Figure 14(a) illustrates that the leakage current consistently stays below 7.6x10-5 A/cm² at voltages above -3.2V. However, at voltages below -3.2V, there is a marked increase in leakage current, culminating in a peak of 0.039A/cm² at -4.0V.

Comment 3

3.Besides the VpL could the authors provide the simulated device bandwidth?

Response 3

Thank you very much for your valuable suggestion on bandwidth simulation, which has significantly expanded the content of the article. We have added the bandwidth simulation of the device and discussed this from lines 384 to 389 on page 12, also illustrating it in Figure 14(b).

The added content is as follows:

An analysis of the device's S11 parameter indicates a -3dB bandwidth of 2.94 GHz, as depicted in Figure 14(b). The utilization of high-k (HK) materials increases overall capacitance compared to traditional SiOâ‚‚ dielectric layers. Although the bandwidth of optical modulators using HK material dielectric layers still necessitates further optimization — especially when contrasted with traditional SiOâ‚‚ layer modulators, which can achieve bandwidths exceeding 10 GHz — this aspect will be a primary focus in our future research endeavors.

Figure 14: Simulation Graphs of Leakage Current and Bandwidth (a) Schematic Illustration of Leakage Current Variation with Voltage (b) Schematic Illustration of Bandwidth based on -S11 Parameter

Round 2

Reviewer 1 Report

Comments and Suggestions for Authors

I am willing to accept the paper in its current form 

Comments on the Quality of English Language

None